# Short-Term Soy Bread Intervention Leads to a Dose-Response Increase in Urinary Isoflavone Metabolites and Satiety in Chronic Pancreatitis

**DOI:** 10.3390/foods12091762

**Published:** 2023-04-24

**Authors:** Jennifer H. Ahn-Jarvis, Daniel Sosh, Erin Lombardo, Gregory B. Lesinski, Darwin L. Conwell, Phil A. Hart, Yael Vodovotz

**Affiliations:** 1College of Food, Agricultural, and Environmental Sciences, Department of Food Science and Technology, The Ohio State University, Columbus, OH 43210, USA; 2College of Public Health, The Ohio State University, Columbus, OH 43210, USA; 3Division of Gastroenterology, Hepatology and Nutrition, The Ohio State University Wexner Medical Center, Columbus, OH 43210, USA

**Keywords:** soy bread, isoflavones, satiety, metabolites, chronic pancreatitis

## Abstract

Patients with chronic pancreatitis (CP) are particularly vulnerable to nutrient malabsorption and undernutrition caused by the underlying pathology of their disease. Dietary intervention trials involving soy isoflavones in patients with CP are limited and isoflavone metabolites have not yet been reported. We hypothesized soy bread containing plant-based protein, dietary fiber, and isoflavones would be well-tolerated and restore gut functional capacity which would lead to isoflavone metabolites profiles like those of healthy populations. Participants (*n* = 9) received 1 week of soy bread in a dose-escalation design (1 to 3 slices/day) or a 4-week maximally tolerated dose (*n* = 1). Dietary adherence, satiety, and palatability were measured. Isoflavone metabolites from 24 h urine collections were quantified using high-performance liquid chromatography. A maximum dose of three slices (99 mg of isoflavones) of soy bread per day was achieved. Short-term exposure to soy bread showed a significant dose-response increase (*p* = 0.007) of total isoflavones and their metabolites in urine. With increasing slices of soy bread, dietary animal protein intake (*p* = 0.009) and perceived thirst (*p* < 0.001) significantly decreased with prolonged satiety (*p* < 0.001). In this study, adherence to short-term intervention with soy bread in CP patients was excellent. Soy isoflavones were reliably delivered. These findings provide the foundation for evaluating a well-characterized soy bread in supporting healthy nutrition and gut function in CP.

## 1. Introduction

Chronic pancreatitis (CP) is a syndrome characterized by chronic inflammation and fibrosis of the pancreas, and is commonly associated with debilitating effects on health outcomes and quality of life [1]. Etiologies include toxin exposures such as alcohol and smoking, recurrent acute pancreatitis, genetics, and idiopathic mechanisms [1]. Chronic inflammation and fibrosis lead to destruction of the parenchyma and progressive loss of endocrine and exocrine function [1]. Consequently, patients often suffer from abdominal pain, diabetes mellitus, and maldigestion. Unfortunately, there are currently no approved medical therapies to interrupt or reverse this process, and management primarily consists of screening for and treating complications [2]. Due to the underlying pathophysiology and functional outcomes of CP, patients are often at risk for malnutrition. Steatorrhea, malabsorption, alcoholism (in alcoholic CP), and poor dietary intake due to abdominal pain and other gastrointestinal symptoms frequently lead to nutrient deficiencies [3,4].

The culmination of maldigestion and malabsorption of nutrients due to exocrine and endocrine insufficiency leads to metabolic perturbations [5], dysbiosis [6,7,8], impaired immunity [9], and loss of intestinal functional capacity [10,11,12] in CP patients. Several studies have reported changes in intestinal microbiota in CP. For instance, a reduction in beneficial bacteria *Faecalibacterium prausnitzii* and *Ruminococcus bromii* was observed in CP subjects compared to healthy controls [7,10,11]. Both bacteria are responsible for production of butyrate, which has a critical role in the growth and differentiation of intestinal epithelial cells as well as supporting healthy immune and intestinal barrier function [13].

Current consensus for the nutritional management of malnutrition in CP patients has been focused on caloric support using high protein, high-energy foods; however, there has been very little attention paid towards recovery of intestinal functional capacity [14]. Soy is rich in proteins (~50% in defatted soy flour) and prebiotics (dietary fiber, linoleic acid, and isoflavones) [15,16] which work synergistically to restore intestinal mucosal integrity, reduce inflammation, improve microbiota composition, and enhance localized production of short chain fatty acids [17,18,19,20]. Specifically, when soy is delivered as a bread versus a beverage, the physicochemical properties of bread prolong the gut transit time and promote the formation of isoflavone metabolites [21,22]. Equol, a mammalian derived metabolite of daidzein, has been purported to display greater bioactivity than its parent compound [23] and it has been associated as a marker of health [24,25]. In previous studies, the frequency of equol-producing phenotypes with those following a Westernized diet [26] was ~30% whereas the frequency doubles to 60% in those consuming an Asian diet [27]. Bacteria belonging to the *Coriobacteriaceae* and *Clostridiaceae* families have been associated with equol-producing phenotypes [24,25,28]. However, the mere presence of these equol-producing bacteria did not ensure equol production; rather, it was a combination of a healthy lifestyle, high microbial diversity, and equol-producing bacteria [29].

Given that metabolic impairments and dysbiosis are commonly found in CP patients, alterations in isoflavone absorption and metabolism are possible. In vitro and animal studies have shown that isoflavones can restore gut functional capacity in gastrointestinal disorders [30,31,32]. Descriptive epidemiology, laboratory animal models, and in vitro studies provide evidence of the health benefits of soy consumption in chronic inflammatory disease; however, data from clinical intervention trials involving whole soy foods in patients with chronic pancreatitis (CP) are almost non-existent. Among the limited number of human studies evaluating the impact of isoflavones on gastrointestinal disease, none have reported isoflavone metabolites in CP [33,34,35]. Therefore, we examined isoflavone absorption and metabolism and dietary habits in an open label trial using soy bread administered for 1 or 4 weeks in subjects with CP. We previously reported that soy bread intervention was safe, associated with a decrease in the pro-inflammatory cytokine tumor necrosis factor alpha (TNFα), and possibly reduced other pro-inflammatory cytokines, such as interferon gamma and interleukin-6 [36]. CP is characterized by a state of chronic inflammation and increased levels of circulating pro-inflammatory cytokines; blunting this process is one purposed mechanism to intervene in the progression of CP [2,9]. Soy bread is a whole-food approach which allows for delivery of high-quality proteins, dietary fiber, and a complex mixture of soy bioactives to target inflammation and nutrients to offset malnutrition in patients with CP.

The current study was a pre-planned secondary analysis assessing the impact of soy bread intervention on isoflavone metabolism and dietary habits in CP participants during dose escalating (DE) and a maximally tolerated dose (MTD) phase. These analyses are needed to determine the potential adherence to soy bread intervention in future participants as well as the potential for inter-subject variability. We hypothesized soy bread containing high-quality plant-based protein, dietary fiber, and isoflavones would be well-tolerated and restore gut functional capacity, which would be evidenced by the presence of isoflavone metabolites like those of a healthy population. The primary objectives of this study were (a) to evaluate tolerability of soy bread intervention in the context of palatability and satiety, and (b) to assess urinary isoflavone absorption and metabolism in CP. Overall, this small clinical trial will provide the necessary clinical data required to move this intervention forward in the setting of a future placebo-controlled trial for individuals with CP. In addition, the findings will provide valuable mechanistic information to understand inflammatory biomarkers relevant to CP and to facilitate in designing a future study of individual differences in soy isoflavone metabolism within this patient population.

## 2. Materials and Methods

### 2.1. Soy Bread Materials

A wheat bread where 30% of the wheat flour was displaced by a mixture of soymilk and soy flour was designed to deliver ~30 mg aglycone equivalents (AE) of total isoflavones/slice. Granulated sugar, kosher salt (Diamond Crystal, Savannah, GA, USA), vegetable shortening (Crisco; Orrville, OH, USA), bread flour (Bouncer; Bay State Milling, Minneapolis, MN, USA), and baker’s yeast (Saf-Instant Yeast; Lesaffre, Marcq-en-Barœul, FR, USA) were purchased from Gordon Food Service (Dublin, OH, USA). Vital wheat gluten (Manildra Milling, Leawood, KS, USA) and defatted soy flour (Baker’s Nutrisoy, Archer Daniel) were purchased from Skidmore Sales (Cincinnati, OH, USA). Soymilk powder (Benesoy; Devansoy, Carroll, IA, USA) was purchased from Devansoy. The standardized soymilk and soy flour mixture was produced using soymilk powder (25 kg) and soy flour (75 kg) that was blended into a single mixture using a large capacity ribbon blender (model 1608, Federal Equipment Co., Cleveland, OH, USA). The soy mix was packaged into 5 kg vacuum sealed bags and stored at −40 °C until used for bread preparation.

### 2.2. Soy Bread Preparation and Isoflavone Analysis

Soy breads were prepared as previously described using a sponge-dough process [37]. Sixty loaves (1300 g each) of bread from 6 batches were manufactured at The Ohio State University Wilbur Gould Food Industries Center over a two-week period. Finished breads were baked until an internal temperature of 96.0 ± 0.5 °C was reached, then cooled at ambient conditions for 4 h, and uniformly sliced (SM302B Doyon, Menominee, MI, USA). Individual slices were weighed and vacuum-sealed in polyethylene bags, then immediately frozen and stored (−40 °C) to extend bread quality and preserve isoflavone content. Frozen breads were distributed to patients in insulated bags supplied with ice packs to keep bread frozen for 2 h. Frozen loaves of bread were distributed with thawing instructions. Isoflavone composition of breads from each batch (*n* = 6) were analyzed using HPLC equipped with photodiode array detector (PDA) (Waters Corp, Milford, MA, USA). The HPLC equipment and conditions as well as preparation of the isoflavone standards are detailed by Ahn-Jarvis et al. [38].

### 2.3. Study Design

The design has been described in detail in our previous publication [36]. In brief, this feasibility study involved two sequential phases: In Phase 1, dose escalation (DE) was used to determine the highest tolerable dose of soy bread. In Phase 2, maximally tolerated dose (MTD) was used to assess tolerance of the maximum target dose during a 4-week intervention (Figure 1). Study subjects were allowed to participate at multiple dose levels.

### 2.4. Participant Description

The study protocol was approved by The Ohio State University Biomedical Science Institutional Review Board (1 October 2014) as protocol 2014H0226 and registered on clinicatrials.gov (NCT 02577640) (https://clinicaltrials.gov/ct2/show/NCT02577640?term=NCT+02577640&draw=2&rank=1 (accessed 23 April 2020). Participants with CP (>18 year of age) who met the eligibility criteria were evaluated, and informed consent was obtained at The Ohio State University Wexner Medical Center Pancreas Clinics. Participants were asked to abstain from legumes (legume-free diet), supplement their diet with the prescribed number of slices, and document the time of soy bread consumption in their daily journal. Condiments were permitted on the bread, but heating was discouraged since the latter may degrade the isoflavones. All participants were provided a standardized multivitamin/mineral supplement (One Daily Multiple Plus Minerals, CVS Caremark, Woonsocket, RI, USA) and asked to discontinue all other dietary supplements (vitamins and botanical supplements) for the duration of the study. Participants were instructed to document gastrointestinal symptoms, frequency of bowel movements, and score their stool using the Bristol stool scale. A registered dietitian provided instructions to complete 3-day diet records (days 2, 4, and 6 for DE and MTD phases as well as additional days −5, −3, and −1 during the MTD phase), questionnaires, and a 24 h urine collection. Fasting blood and spot urine were collected at the start of intervention (baseline) and at the end of the 7-day intervention for DE and every week for the MTD participant.

### 2.5. Sensory and Satiety Assessments

Participants were provided with a sensory ballot with written instructions to complete at home. They were to be completed on the first day, immediately after bread was consumed. Participants were instructed to rinse their mouths with cold tap water prior to eating their bread. A 9-point hedonic scale (1 = dislike extremely to 9 = like extremely) was used to assess six attributes (overall liking, aroma, flavor, sweetness, bitterness, and texture) of the soy bread and was followed by a descriptive analysis. It was performed using a horizontal line scale (6 inches) to evaluate whether soy bread attributes were like their ideal bread. Intensity was measured (1 = low and 10 = high) for brown crust color, sweetness, saltiness, bitterness, hardness, chewiness, and oiliness. Participants were given the following written lexicons to calibrate their senses to the respective attribute: peanut (1) and milk chocolate (10) for brown color; potato (1) and carrot (10) for sweetness; tomato (1) and olives (10) for saltiness; spinach (1) and coffee (10) for bitterness; cooked pasta (1) and licorice candy (10) for hardness; cream cheese (1) and raisin (10) for chewiness; nuts (1) and French fries (10) for oiliness. Additionally, measures of satiety (perceived fullness or lack of hunger) were conducted using a vertical (6 inches) visual analogue scale (VAS) to examine the impact of satiety on soy bread compliance. Subjects were asked to rate their satiety (hunger, fullness, how much they can eat, thought of food, desire, and urge to eat) as well rate their nausea, thirst, and energy. VAS was to be completed prior to the first dose of soy bread consumed, then 30 min and 4 h after consumption of the soy bread, as well as before going to bed. For the first day, participants were asked to consume a full dose as a single meal.

### 2.6. Laboratory and Clinical Measures of Toxicity

Fasting blood (electrolytes, glucose, renal and liver function tests) was collected at the enrollment and end of study visits. All clinical blood samples were analyzed by The Ohio State University Wexner Medical Center clinical laboratories.

### 2.7. Isoflavone Quantification in Urine

#### 2.7.1. Chemicals

From Fisher Scientific (Fairlawn, OH, USA), anhydrous sodium acetate and solvents (diethyl ether, formic acid, acetonitrile, methanol, and water) which were HPLC-grade or higher were purchased. Lyophilized glucuronidase/sulfatase from Helix pomatia, dimethyl sulfoxide, and equol were obtained from Sigma-Aldrich (St. Louis, MO, USA). Daidzein, genistein, glycitein, daidzin, genistin, glycitin, acetyl daidzin, malonyl genistin, and acetyl genistin were obtained from LC laboratories division (PKC Pharmaceuticals, Inc., Woburn, MA, USA), and malonyl daidzin and malonyl glycitin from Wako Chemicals USA, Inc., (Richmond, VA, USA). Isoflavone metabolites (dihydrodaidzein DHD, dihydrogenistein DHG, ODMA, and 6-OH-ODMA) were acquired from Plantech (Reading, UK). The following authenticated isoflavone standards were purchased from Sigma-Aldrich (St. Louis, MO, USA): daidzein, genistein, glycitein, and equol. Food isoflavone standards (daidzin, genistin, glycitin, acetyl daidzin, malonyl genistin, and acetyl genistin) were obtained from LC Laboratories division (PKC Pharmaceuticals, Inc., Woburn, MA, USA). 

#### 2.7.2. Sample Handling and Isoflavone Quantification

Isoflavones were quantified using high-performance liquid chromatography (HPLC) grade chemicals and authenticated isoflavone standards as described by Ahn-Jarvis et al. 2015 [37]. Subjects were provided 24 h urine collection vessels, which were pre-weighed with 0.5 g/L boric acid, and collection began one day prior to completion of study (i.e., Day 6 and/or Day 27). Mass of urine combined with urine specific gravity was used to determine urine volume over 24 h. A minimum collection of 20 h was considered complete and used for analysis. Urine aliquots were stored at −80 °C for HPLC analysis. Extraction and quantification of isoflavonoids in urine are detailed by Ahn-Jarvis et al. [37]. Regression analysis of calibration curves revealed a good linear relationship (R^2^ = 0.9996 ± 0.0002). The inter-assay and intra-assay variability in urine was less than 15%. Isoflavone concentrations in urine less than 43 nmol/L (12.2 ng/mL) to 54 nmol/L (13.7 ng/mL) for the parent (daidzein, genistein, glycitein) isoflavonoids and less than 48 nmol/L (13.1 ng/mL) to 62 nmol/L (15.9 ng/mL) for intermediate (DHG and DHD) isoflavonoids were considered below the level of quantification. Microbial metabolites of isoflavones less than 120 nmol/L (29.1 ng/mL) for equol, 88 nmol/L (22.7 ng/mL) for O-DMA, and 137 nmol/L (39.5 ng/mL) for 6 OH-ODMA were considered below the level of quantification.

### 2.8. Statistical Analysis

SPSS software (version 22, IBM, Armonk, NY, USA) was used for analyses. A paired t-test was used to evaluate the differences between baseline and at the end of study in clinical biomarkers. An ANOVA was used to discriminate differences in satiety parameters, isoflavone concentrations in urine, and dietary intake in respects to dose (slices/day) and/or time. When statistically significant (*p* < 0.05) differences were found, a Tukey’s post hoc test was used to determine the mean separation. Hierarchical cluster analysis for urine isoflavones was conducted using proportions of metabolites belonging to the daidzein family as three components (equol, ODMA, and DHD + Daidzein). The clusters were formed using Euclidean distances and average linkage. These methods are detailed in our previous studies [37,39]. Sensory data were analyzed using Wilcoxon signed-rank test and reported as median ± inter-quartile range. Radar analysis compared soy bread attributes to participants’ self-described ideal bread using paired *t*-test.

## 3. Results and Discussion

### 3.1. Soy Bread Nutrient and Isoflavone Composition

Each slice of soy bread provided ~34 mg total isoflavones. The nutrient and isoflavone composition of soy bread are detailed in Table 1. Soy bread isoflavone composition remained stable during 2 years of frozen storage.

### 3.2. Soy Bread Palatability and Its Impact on Satiety

Sensory evaluation of soy bread using hedonic scoring indicated that the soy bread was palatable (Figure 2A). The radar analysis assessing bread attributes indicated that the taste of the soy bread was like their ideal bread, but the texture (hardness and chewiness) of the soy bread was significantly different (*p* = 0.040 and *p* = 0.003, respectively, Figure 2B) which is evidenced by a lower Hedonic score (likability) for texture being below neutral. Because of the 30% displacement of wheat flour by soy ingredients, CP participants detected a difference in the texture of the soy bread not being the same as their ideal bread and this was also reflected in the decrease in Hedonic score for texture. Loss in organoleptic qualities of wheat bread have been reported when 15% or 20% of the wheat flour is displaced by soy [40]. Moreover, there were no significant dose-dependent effects on bread palatability among the three cohorts nor the MTD participant. Therefore, consistent with our primary hypothesis, the sensory evaluation of the soy bread suggested that the bread would be well-tolerated and feasible for use in a short-term intervention study.

Visual analogue scale (VAS) was used to evaluate various aspects of satiety. The VAS scores indicate perceived thirst decreased significantly (*p* < 0.001) between one and three slices of bread after 8 h (Figure 2C). Dietary intake records show that dietary water and total beverage intake did not significantly decrease (*p* = 0.065 and *p* = 0.152, respectively) but there was a trend where mean dietary water intake decreased with increasing number of slices/day. Further, VAS scores indicate that there was a significant persistence in satiety with an increase in number of daily slices over time. Specifically, in the assessment of desire to eat, one slice/day compared to three slices/day at 8 h postprandial (*p* < 0.001) showed a decrease in the desire to eat (Figure 2B). Similar differences were observed with other satiety endpoints (Appendix A). The significant differences in perceived thirst and satiety suggest that soy bread may be contributing to fluid homeostasis in the gut. Although speculative, one possible rationale is the physicochemical differences between wheat and soy flour. Soy flour has three times the water-holding capacity than wheat flour and strong oil holding capacity [41,42]. This physicochemical property may facilitate water retention in the intestinal lumen, thereby enhancing satiety and fluid balance, but further dietary intervention studies using wheat control bread is warranted.

### 3.3. Participant Clinical Features and Dietary Adhrence

Recruitment occurred over a 20-month period and a detailed CONSORT diagram was previously reported [36]. Of the 11 participants enrolled, 1 failed screening and withdrew before allocation, 9 completed the DE phase and 1 completed the MTD phase. Among the nine, two individuals from cohort 1 (1 slice/day) returned after 1 year to participate in cohort 3 (3 slices/day) (Figure 1). The age range was from 52 to 79 years old, and all were men except for one individual. A majority consumed their meals with lipase (Pancreatic Enzyme Replacement Therapy, PERT) supplementation (Table 2). Fasting blood chemistry, complete blood count, and renal and liver function tests were not significantly affected by soy bread intervention (Table 2). Although not a primary outcome in this small study, we observed a 18% reduction in mean fasting blood glucose in cohorts consuming two and three slices of soy bread per day. Previous animal studies report that isoflavones elicit hypoglycemic effects in rats with diabetes by inhibiting intestinal α-glucosidase activity [43] whereas others report that isoflavones improve insulin sensitivity by activating peroxisome proliferator activated receptors [44,45]. However, the anti-diabetic effects of soy isoflavones in human studies are mixed [46,47].

From three-day diet records, the total energy intake (*p* = 0.095) was not statistically significant with increases in soy bread dose, but this is likely attributed to the small size of this study. However, we observed a 9% and 35% reduction in mean energy intake in cohort 2 and 3 when compared to cohort 1. Likewise, there was a decrease, albeit not statistically significant, in daily water intake by ~35% in cohort 2 and 3 when compared to cohort 1. The dietary intake of total protein (*p* = 0.013), specifically animal protein (*p* = 0.009) decreased in a dose dependent manner (Table 2). Dietary intake was determined without the inclusion of soy bread. Participants were instructed to maintain their normal diets and consume breads in addition to their normal diets. Notably, the caloric content of the soy bread (weight for weight) is approximately 10% lower than a slice of conventional white or whole wheat bread. In a recent study, we demonstrated there is a wide range of caloric intake in subjects with CP, so this possible shift could have mixed consequences [48]. For those with substantial calories derived from lower quality food items using this bread product would be beneficial, whereas intake could be problematic in those with low caloric intake at baseline. Additionally, it will be necessary to assess tolerability over a long term, since it is anticipated that this (or any) intervention to effectively treat CP would need to last for years.

### 3.4. Isoflavone Metabolism

Oral ingestion of soy bread, when delivered in a dose escalating manner by increasing the number of bread slices per day, was associated with a linear increase (Pearson R = 0.869, *p* = 0.0011) in the excretion of total isoflavones in 24 h urine collections (Figure 3C). The total isoflavone excreted in each cohort during the DE phase was statistically significant, *p* = 0.007. The mean (±SEM) total daidzein excretion was 11.50 ± 3.55 mg/24 h for one slice, 19.02 ± 5.00 mg/24 h for two slices, and 30.51 ± 5.10 mg/24 h for three slices (Figure 3A), while total genistein was 4.66 ± 1.47 mg/24 h for one slice, 11.55 ± 3.05 mg/24 h for two slices, and 15.91 ± 2.95 mg/24 h for three slices (Figure 3B). A significant dose-dependent increase was observed in total daidzein (*p* = 0.028) excreted, but not for total genistein (*p* = 0.188). Daidzein metabolites such as dihydrodaidzein (DHD), O-desmethyl-angolensin (ODMA), and equol and genistein metabolites such as dihydrogenistein (DHG) and 6-OH-ODMA) were observed in 24 h urine samples.

### 3.5. Isoflavonoid Phenotypes

Phenotypes which favor daidzein excretion were predominant compared to genistein excreting phenotypes. Therefore, in this study population, 90% (9/10) had total daidzein: total genistein ratios (TD:TG ratio) greater than 1 (range 1.28–9.21). Like our previous soy bread studies [21,37], we observed four daidzein metabolizing phenotypes with this CP cohort. Specifically, among the ten 24 h urine samples analyzed, we observed one non-daidzein metabolizing (NDM) phenotype (subject 1G), three O-desmethyl-angolensin (ODMA: subjects 1D, 1E, and 2A), three ODMA + equol (subjects 1A, 1C, and 1H), and three equol (subjects 1B, 1F, and 1I) producing phenotypes (Figure 3A). The greatest excretion of total genistein was observed in NDM phenotype (subject 1G). Two subjects repeated 1 and 3 slice/day dosing arms. Both demonstrated that isoflavone metabolizing phenotypes are reproducible and metabolite excretion increased with dose. In the equol-producing subject (1B and 1I) equol excretion increased from 28 (1 slice/day) to 61% (3 slice/day) of total isoflavones excreted, whereas in a ODMA producing subject (1A and 1H) their ODMA excretion increased from 3% (1 slice/day) to 10% (3 slice/day). In this same participant, equol was not present during their 1 slice/day period but detected during their 3 slice/day intervention period.

Isoflavone absorption and metabolism patterns in our cohort were similar with those consuming a habitual Asian diet [49] and those consuming a Westernized diet after soy intervention [21,37]. Repeated daily dosing in both 1-week and 4-week interventions demonstrated both time periods were sufficient in duration to capture isoflavone metabolites in urine. Isoflavone microbial metabolites have shown greater bioactivity than their parent compounds (daidzein, genistein, and glycitein) [50] and their production shows great interspecies and regional diversity. Among mammalian species, the frequency of equol production is much higher in rodents than in humans, more frequent in Asian than Western populations, and much higher in vegetarian diets [49,51]. In our cohort, equol-producing compared to NDM and ODMA phenotypes had the greatest daily intake of total dietary fiber (*p* = 0.001) and insoluble fiber (*p* = 0.002). The NDM (subject 1G) phenotype had the lowest caloric intake at ~800 kcal/day whereas the mean daily energy intake for the other phenotypes was ~2000 kcal/day.

### 3.6. Potential Implications for Clinical Care

CP is a debilitating, irreversible disease with a detrimental impact on quality of life, nutrition, and health care visits. While there are currently no approved medical therapies to interrupt this chronic inflammatory process, much of the nutritional management of chronic pancreatitis is to prevent malnutrition by treating the symptoms of maldigestion and malabsorption using medical modalities and diet modification (high protein, high energy diet) [14,52]. However, this fails to address a number of the underlying pathways that contribute to disease progression and clinical manifestations, including inflammatory pathways and gut dysbiosis, amongst others. Advances in microbiome research and whole genome sequencing have provided insights into other alternative approaches, such as restoration of intestinal functional capacity and gut health to improve the nutritional status in CP, and warrant further investigation in CP [53,54,55]. Among the many food ingredients, plant-based foods offer a unique opportunity to enhance bioactive profiles in the food ingredients through horticultural conditions, biofortification, and precision breeding [56,57,58,59]. Moreover, complementary food processing techniques can enhance bioavailability of bioactive compounds or utilize food structures to modulate their delivery to localized sites along the gastrointestinal tract [38,60]. While nutritional intervention studies represent a great opportunity in CP, there are several challenges, including the lack of standardized outcome assessments to determine clinical efficacy [61]. Nevertheless, considering the absence of approved medical therapies for CP and the significant nutritional abnormalities, this is a key opportunity for future investigations.

The current pilot trial was primarily focused on assessing feasibility, safety, and tolerability of the soy-enriched bread in a study population with chronic pancreatitis. Due to the small sample size and expected within-group variability, we were unable to characterize the mechanisms of cellular action or immunological signaling, and examined isoflavone metabolism as a biochemical outcome. Nevertheless, a prior pre-clinical resource using a similar diet in an animal model demonstrated that dietary soy modulates natural killer cell function, including reduced expression of interferon-gamma, a key mediator for chronic pancreatitis [62].

## 4. Conclusions

In this small pilot study, short-term soy intervention with soy bread in participants with CP demonstrated that the soy bread effectively delivered isoflavones in a dose-dependent manner. To our knowledge, this study is the first to report isoflavone metabolism and isoflavone phenotypes in patients with chronic pancreatitis. Because of the great heterogeneity in CP disease, metabolic phenotypes of soy isoflavone intervention are critical to help decipher heterogeneity in biologic responses among individuals with CP. This study combined with our previous findings demonstrating safety and the anti-inflammatory effects of the soy bread provides the necessary foundation to design future foods for CP and larger clinical trials.

## Figures and Tables

**Figure 1 foods-12-01762-f001:**
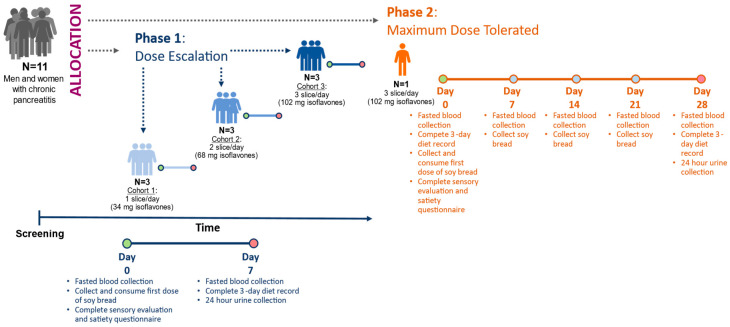
Study design of 3 + 3 dose escalating phase leading to maximum tolerated dose phase. Depending on the study phase, isoflavones were measured in urine on day 7 or day 28. Circles represent study visit days.

**Figure 2 foods-12-01762-f002:**
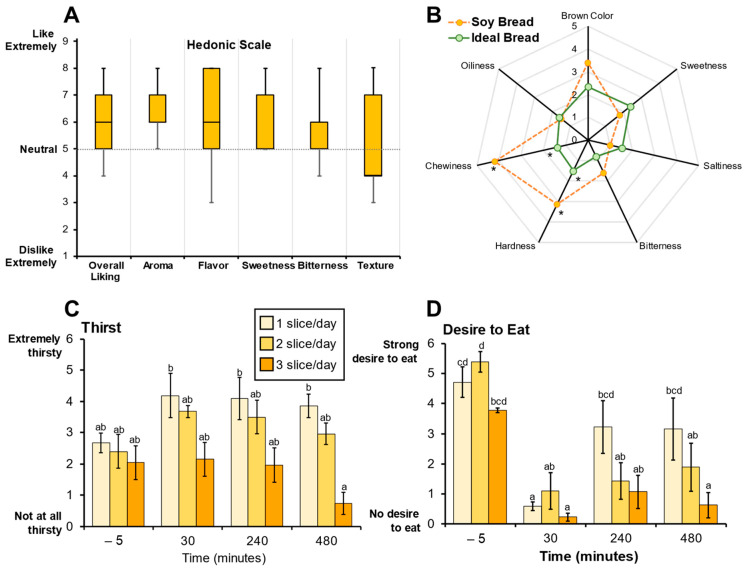
Sensory evaluations were used to assess soy bread palatability. (**A**) Hedonic scores representing soy bread palatability shown as box and whisker plots (median and interquartile ranges). (**B**) Radar analysis shows attribute intensities (median) of soy bread compared to participant’s ideal bread. Visual Analogue Scale (VAS) scores for perceived thirst (**C**) and desire to eat (**D**) are represented as bar graphs (mean ± SD). Superscript letters represent the mean separation using ANOVA analysis and Tukey’s post hoc test when significant differences (*p* ≤ 0.05) were observed. * Significant differences (*p* ≤ 0.05) between soy bread and ideal bread were assessed using a paired *t*-test.

**Figure 3 foods-12-01762-f003:**
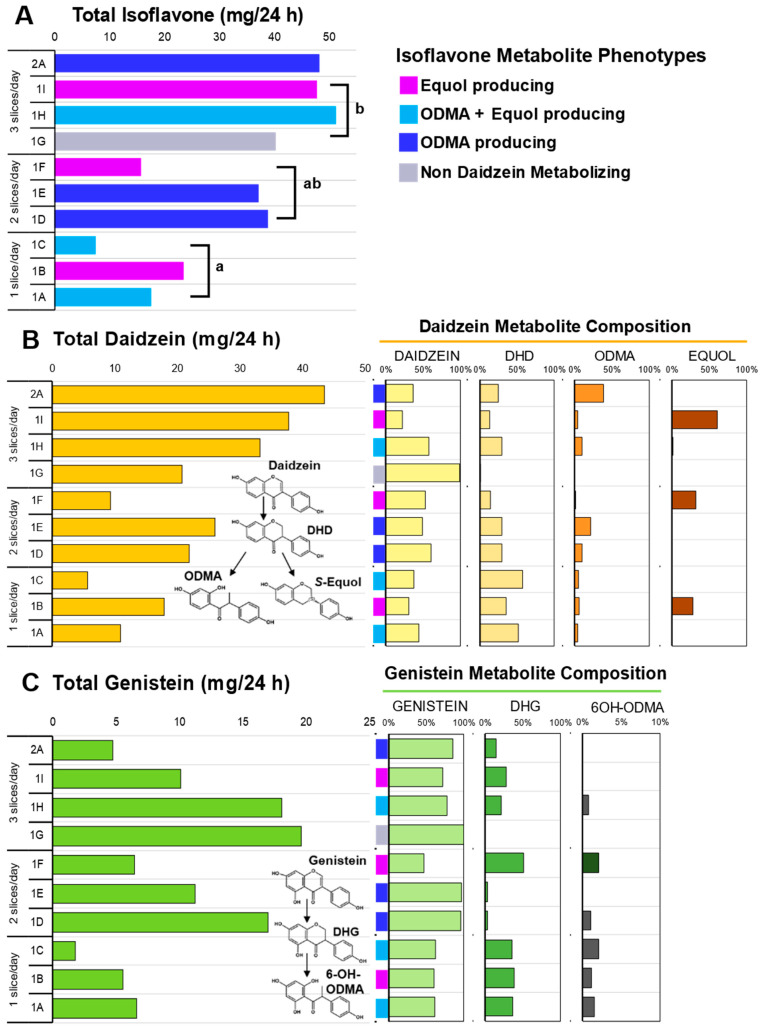
Isoflavone metabolites in 24 h urine collection across three doses of soy bread. (**A**) Total isoflavones excreted across three doses of soy bread and colors represent the four daidzein excreting phenotypes observed in this cohort. (**B**) Total daidzein concentration excreted (**left**) and their corresponding metabolite composition of each subject expressed as a percentage of total daidzein (**right**). (**C**) Total genistein concentration excreted (**left**) and their corresponding metabolite composition of each subject expressed as a percentage of total genistein (**right**). Superscript letters represent the mean separation using ANOVA analysis and Tukey’s posthoc test when significant differences (*p* ≤ 0.05) were observed. Isoflavone structures show abbreviated metabolite pathways (inset). DHD: dihydrodaidzein; DHG: dihydrogenistein; ODMA: O-desmethyl-angolensin; 6-OH-ODMA: 6-hydroxyl ODMA.

**Table 1 foods-12-01762-t001:** Nutrient and isoflavone content of one slice of soy bread.

Nutrition Parameters	
Serving Size (g)	92.4 ± 2.3
**Nutrient Composition** ^1^	
Energy (kcal)	220.8
Fat (g)	3.4
Total Carbohydrates (g)	37.8
Dietary Fiber (g)	3.7
Protein (g)	15.2
**Total Isoflavones** (Mean ± SD, mg) ^2^	**34.38 ± 0.21**
Daidzein (mg)	1.44 ± 0.10
Daidzin (mg)	3.81 ± 0.23
6″-O-Acetyldaidzin (mg)	0.54 ± 0.03
6″-O-Malonyldaidzin (mg)	6.48 ± 0.39
Genistein (mg)	1.84 ± 0.14
Genistin (mg)	6.92 ± 0.40
6″-O-Acetylgenistin (mg)	0.67 ± 0.03
6″-O-Malonylgenistin (mg)	11.67 ± 0.63
Glycitein (mg)	0.10 ± 0.01
Glycitin (mg)	0.77 ± 0.04
6″-O-Acetylglycitin (mg)	Trivial ^3^
6″-O-Malonylglycitin (mg)	0.12 ± 0.01

^1^—Nutrient calculated from soy bread formulation using NDSR; ^2^—Isoflavone content in soy breads (*n* = 6) determined using HPLC with PDA; ^3^—Quantity less than level of quantification but detectable.

**Table 2 foods-12-01762-t002:** Clinical and dietary information of chronic pancreatitis participants.

Participant Information	Dose Escalating Phase (Phase 1)			MTD ^1^ (Phase 2)
	1 Slice/Day (*n* = 3)	2 Slice/Day (*n* = 3)	3 Slice/Day (*n* = 3) ^2^	3 Slice/Day (*n* = 1)
**Clinical Features**				
Body Mass Index (kg/m^2^,mean ± SD)	23.4 ± 5.2	28.4 ± 1.6	26.5 ± 3.9	24.6
Lipase (PERT) with meals (%)	66	100	66	yes
Lipase dose range (1000 units/meal)	24 and 72	24 to 48	24 and 72	24
Diabetes mellitus (%)	33	100	66	no
Pancreatic calcification/calculi (%)	100	100	100	yes
Main Pancreatic Duct Dilatation (%)	66	66	100	no
**Clinical Laboratory Values** (mean ± SD) ^3^				
Creatinine (mg/dL)				
Pre-intervention	1.03 ± 0.06	1.17 ± 0.02	0.94 ± 0.09	1.49
Post-intervention	0.09 ± 0.06	1.24 ± 0.05	0.89 ± 0.07	1.49
Blood glucose (mg/dL)				
Pre-intervention	104 ± 6	172 ± 52	236 ± 141	82
Post-intervention	113 ± 8	139 ± 24	194 ± 79	89
Alanine aminotransferase (IU/L)				
Pre-intervention	16 ± 5	22 ± 10	17 ± 2	12
Post-intervention	18 ± 4	26 ± 10	17 ± 3	16
Aspartate aminotransferase (IU/L)				
Pre-intervention	16 ± 1	18 ± 4	17 ± 1	19
Post-intervention	22 ± 4	19 ± 4	18 ± 3	21
Total bilirubin (mg/dL)				
Pre-intervention	0.6 ± 0.1	0.4 ± 0.1	0.7 ± 0.1	0.4
Post-intervention	0.5 ± 0.1	0.4 ± 0.1	0.7 ± 0.1	0.4
Albumin (g/dL)				
Pre-intervention	4.3 ± 0.2	4.3 ± 0.3	4.1 ± 0.2	3.6
Post-intervention	4.3 ± 0.2	4.3 ± 0.3	4.1 ± 0.2	3.6
Alkaline phosphatase (IU/L)				
Pre-intervention	74 ± 12	110 ± 25	101 ± 45	98
Post-intervention	70 ± 7	109 ± 16	108 ± 34	101
**Daily Nutrient Intake** (mean ± SEM) ^4^				
Total Energy (kcal)	2300 ± 288	2095 ± 240	1517 ± 221	1592 ± 158
Total fat (g)	95.0 ± 16.8	75.2 ± 11.7	65.7 ± 14.4	49.0 ± 18.6
Cholesterol (mg)	391.4 ± 109.5	174.3 ± 33.2	216.9 ± 64.5	81.6 ± 71.1
Total carbohydrate (g)	278.1 ± 32.2	288.8 ± 38.5	191.5 ± 31.8	252.7 ± 46.3
Dietary fiber (g)	21.4 ± 5.6	18.4 ± 2.9	25.6 ± 5.5	24.9 ± 11.3
Total protein(g) ^5^	96.7 ± 13.4 ^a^	74.6 ± 8.3 ^ab^	48.2 ± 9.6 ^b^	50.1 ± 12.1
Animal protein (g) ^5^	74.7 ± 11.3 ^a^	46.4 ± 9.6 ^ab^	28.6 ± 7.7 ^b^	18.5 ± 12.6
Vegetable protein (g)	22.0 ± 3.6	28.2 ± 3.3	19.6 ± 3.4	29.1 ± 8.2
Total water (g)	3296 ± 936	2149 ± 517	2083 ± 514	2143 ± 189

^1^—Maximum Tolerated Dose; ^2^—2 of 3 subjects were also in the 1 slice/day subgroup; ^3^—Fasting blood serum; ^4^—Data collected from 3-day diet records, not including intake of soy bread, and analyzed using NDSR (Nutrition Data System for Research, Minnesota, MN, USA); ^5^—Statistically significant (*p* < 0.05) ANOVA analysis and mean separation are indicated by different superscript letters.

## Data Availability

All related data and methods are presented in this paper. Additional inquiries should be addressed to the corresponding author.

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
