# Peer review of "Short-Term Soy Bread Intervention Leads to a Dose-Response Increase in Urinary Isoflavone Metabolites and Satiety in Chronic Pancreatitis"

_foods, 2023, doi:10.3390/foods12091762_

Round 1
Reviewer 1 Report
This is a high quality paper, scientifically sound and clearly written. There are only a few minor suggested corrections which could improve it. Their list follows.
Line 60 enhance
Line 62 promote
Line 163 The listed attributes are six but they are described as being seven! please check
Line 187 cancel "that"
Line 237 Table 1. In the table footnote, please use asterisks instead of numbers to match the symbols in the table
Line 254 intensities
Line 278 What happened to the 11th? Please describe
Line 287 "another report.......": here there is something wrong either with the English Grammar correspondence (reports) or with the word "another" because there are 2 references mentioned.
Line 364 repeated? instead of repeat
Line 407 cancel "&"
Author Response
Reviewer 1:
This is a high quality paper, scientifically sound and clearly written. There are only a few minor suggested corrections which could improve it. Their list follows.
Thank you!
Line 60 enhance
In Line 61* the term “enhances” has been replaced with “enhance” *Note all line numbers refer to the clean copy version document”
Line 62 promote
In line 63 the term “promotes” has been replaced with “promote”
Line 163 The listed attributes are six but they are described as being seven! please check
Thank you for your careful review. In line 171, seven has been replaced by six.
Line 187 cancel "that"
In line 195, the term “that” has been removed.
Line 237 Table 1. In the table footnote, please use asterisks instead of numbers to match the symbols in the table
In Table 1 (line 245-246), the asterisks in the table have been replaced with numbers to align with the author guidelines for the journal.
Line 254 intensities
In Figure 2’s legend (line 264), the typographical error was corrected and the term “intenstities” has been replaced with “intensities”.
Line 278 What happened to the 11th? Please describe.
Lines 287 to 288 have been corrected to account for all 11 participants enrolled in the study. The corrected sentence reads as such, “Of the 11 participants enrolled, 1 failed screening and withdrawn before allocation, 9 completed the DE phase and 1 completed the MTD phase”
Line 287 "another report.......": here there is something wrong either with the English Grammar correspondence (reports) or with the word "another" because there are 2 references mentioned.
In lines 297 to 299, the grammar has been corrected and the revised sentence reads as such, Previous animal studies report that isoflavones elicit hypoglycemic effects in rats with diabetes by inhibiting intestinal α-glucosidase activity [43] whereas others report that isoflavones improve insulin sensitivity by activating peroxisome proliferator activated receptors [44,45].
Line 364 repeated? instead of repeat
Thank you for the comment but the sentence with this grammatical error was removed to address the comments of reviewer 2.
Line 407 cancel "&"
In line 424, the ampersand (&) has been removed.

Reviewer 2 Report
After a careful review of the manuscript entitled “Short-term soy bread intervention leads to a dose-response increase in urinary isoflavone metabolites and satiety in chronic pancreatitis” here are few minor general recommendations. Overall, it is a well written and informative review article, but I suggest some revisions in the article. Please address the following points in the revised manuscript.
1. Please add one are more sentences about the novelty as it is not given.
2. Objectives of the study should be clearly mentioned in the manuscript with more details.
3. In the abstract, also mention the major findings and best one results.
4. In the introduction, a lot of grammar and sentences mistakes are observed, try to remove these.
5. Revise the format of the Tables given, Follow the author guide lines for figures and tables in the manuscript.
6. The figures should be changed as these are difficult to understand.
7. The result needs more details discussion regarding the trend and complete mechanism.
8. At the last, conclusion should be short and relevant to the study.
Author Response
Reviewer 2:
After a careful review of the manuscript entitled “Short-term soy bread intervention leads to a dose-response increase in urinary isoflavone metabolites and satiety in chronic pancreatitis” here are few minor general recommendations. Overall, it is a well written and informative review article, but I suggest some revisions in the article. Please address the following points in the revised manuscript.
1. Please add one are more sentences about the novelty as it is not given.
To add further clarity about the novelty of our study, we have added to and/or modified the abstract and introduction.
-
- In the abstract (lines 18 to 19*), we have added the following sentence: “Dietary intervention trials involving soy isoflavones in patients with CP are limited and isoflavone metabolites have not yet been reported.” *Note all line numbers refer to the clean copy version document.”
- In the introduction (lines 77 to 79) the following sentence was added “Descriptive epidemiology, laboratory animal models, and in vitro studies provide evidence of the health benefits of soy consumption in chronic inflammatory disease; yet data from clinical intervention trials involving whole soy foods in patients with chronic pancreatitis (CP) is almost non-existent.” and the original sentence “However, human studies evaluating the impact of iso-flavones on gastrointestinal disease, are limited and most often the isoflavone metabolites have not been reported in these cohorts [33-35]” has been replaced with “Among the limited number of human studies evaluating the impact of isoflavones on gastrointestinal disease, none have reported isoflavone metabolites in CP [33-35]”
2. Objectives of the study should be clearly mentioned in the manuscript with more details.
Thank you for this comment, we have added the following sentence to lines 99 to 101: “The primary objectives of this study were a.) to evaluate tolerability of soy bread intervention in the context of palatability and satiety b.) to assess urinary isoflavone absorption and metabolism in CP.”
3. In the abstract, also mention the major findings and best one results.
In the abstract, we have mentioned the following major findings: “Short-term exposure to soy bread showed a significant dose-response increase (p=0.007) of total isoflavones and their metabolites in urine. With increasing slices of soy bread, dietary animal protein intake (p=0.009) and perceived thirst (p<0.001) significantly decreased with prolonged satiety (p< 0.001). In this study, adherence to short-term intervention with soy bread in CP patients was excellent.”
4. In the introduction, a lot of grammar and sentences mistakes are observed, try to remove these.
The entire manuscript was reviewed for grammatical errors, corrected, and the specific changes can be found in the document version with tracked changes. We are amenable to any further recommended changes during copy proofing to fit the stylistic preferences of the journal.
5. Revise the format of the Tables given, Follow the author guide lines for figures and tables in the manuscript.
In Table 1 and 2 the footnotes have been reformatted to align with those provided in the author guidelines. Therefore, the asterisks in Table 1 and the cross in Table 2 have been replaced by numerals. Moreover, a pdf of manuscript has been submitted to ensure formatting remains unchanged with submission.
6. The figures should be changed as these are difficult to understand.
Figure 1: the study design depiction has been changed for clarity and details of study activities have been added. Additionally, due to the change to figure 1, the graphical abstract figure has been revised.
Figure 2: major changes were made to the corresponding text (lines 253 to 260) and a figure legend has been included.
-
- Results and Discussion section 3.2 (lines 253 to 260) revised sentences reads as such: Because of the 30% displacement of wheat flour by soy ingredients, CP participants detected a difference in the texture of the soy bread not being the same as their ideal bread and this was also reflected in the decrease in Hedonic score for texture. Loss in organoleptic qualities of wheat bread have been reported when 15 or 20% of the wheat flour is displaced by soy [40]. Moreover, there were no significant dose-dependent effects on bread palatability among the three cohorts nor the MTD participant. Therefore, consistent with our primary hypothesis, the sensory evaluation of the soy bread suggested that the bread would be well-tolerated and feasible for use in a short-term intervention study.
- Figure 2 legend (lines 261to 268) has been changed to: Sensory evaluations were used to assess soy bread palatability. A, Hedonic scores representing soy bread palatability shown as box and whisker plots (median and interquartile ranges). B, Radar analysis shows attribute intensities (median) of soy bread compared to participant’s ideal bread. Visual Analogue Scale (VAS) scores for perceived thirst (C) and desire to eat are represented (D) as bar graphs (mean ± SD). Superscript letters represent the mean separation using ANOVA analysis and Tukey’s post hoc test when significant differences (p≤0.05) were observed. *Significant differences (p≤0.05) between soy bread and ideal bread were assessed using a paired t-test.
Figure 3: this figure was modified to illustrate concentrations of total isoflavone, total daidzein, and total genistein excreted from their corresponding compositions (percentage of daidzein or genistein) and phenotype designations. Moreover, additional information was added to the figure legend to improve the clarity between the concentration of isoflavones (daidzein and genistein) and their corresponding metabolite composition.
7. The result needs more details discussion regarding the trend and complete mechanism.
We appreciate the reviewers request for additional information regarding the potential mechanism of benefit from soy isoflavone supplementation for patients with chronic pancreatitis. We have provided additional information regarding a potential mechanism and have included this at the end of the Results and Discussion section of the revised manuscript (lines 375 to 403).
8. At the last, conclusion should be short and relevant to the study.
To address this recommendation, we have inserted an additional subsection into the Results and Discussion section to allow us to simply focus on a high level summary of the study in the Conclusion section.
